# Prevalence and risk factors of hypertension among civil servants in Sidama Zone, south Ethiopia

**Bedilu Badego**[1]*, **Amanuel Yoseph**[2], **Ayalew Astatkie**[2]

**1** Sidama Zone Health Department, Hawassa, Ethiopia, **2** School of Public Health, College of Medicine and Health Sciences, Hawassa University, Hawassa, Ethiopia

* baynbadego@gmail.com

**Data Availability Statement:** All relevant data are within the paper and its Supporting Information files.

**Funding:** The author(s) received no specific funding for this work.

## Abstract

### Introduction

Hypertension is the leading cause of death and disability in adult populations globally. Its prevalence is increasing rapidly in Ethiopia. Studies conducted to date address different population categories. However, there is lack of data on the prevalence and risk factors of hypertension among civil servants working in various sectors and levels.

### Objective

To assess the prevalence and risk factors of hypertension among civil servants in Sidama Zone, south Ethiopia.

### Methods and materials

An institution-based cross-sectional study was conducted from March 1–30, 2019 on a sample of 546 civil servants selected randomly from different departments of Sidama Zone Administration. Data were collected using structured, face-to-face interviewer-administered questionnaire and standard physical measurements. The data were entered using Epi Data 3.1 and analyzed using SPSS version 20. Multivariable logistic regression analysis was used to identify factors associated with hypertension. Adjusted odds ratios (AORs) with 95% confidence interval (CI) were computed to assess the presence and strength of associations.

### Results

A total of 546 civil servants responded resulting in a response rate of 94.9%. The prevalence of hypertension was 24.5% [95% CI: 23.3% - 25.6%]. The identified risk factors of hypertension were male sex (AOR 4.31[95% CI: 1.84–10.09]), moderate current alcohol consumption (AOR: 4.85; [95% CI: 1.73–13.61]), current khat chewing (AOR 2.97[95% CI: 1.38–6.40]), old age (AOR: 4.41[95% CI: 1.19–16.26]), being obese (AOR 5.94 [95% CI: 1.26–27.86]) and central obesity (AOR 3.57 [95% CI: 1.80–7.07]).

### Conclusions

One in four civil servants are hypertensive. Different demographic, behavioral and metabolic factors increase the odds of hypertension among civil servants. Prevention and control of

**Competing interests:** The authors have declared that no competing interests exist.

hypertension shall involve promotion of healthy lifestyles such as weight management, regular physical activity and quitting or cutting down on harmful use of substances such as alcohol and khat.

## Introduction

Non-communicable diseases (NCDs) are the major causes of morbidity and premature deaths under the age of 70 years globally. Nearly half of NCD deaths occur due to cardiovascular diseases (CVDs). Elevated blood pressure is a major risk factor for CVDs [1].

According to the Global Burden of Diseases report, hypertension is the leading cause of death and disability among adults globally [2]. In adults older than 25 years, approximately 4 in 10 individuals are hypertensive and nowadays hypertension is a public health epidemic across the world [3]. Hypertension is also a leading risk for dementia, renal failure, and fetal and maternal death in pregnancy [4, 5].

It is estimated that globally 1.13 billion people are hypertensive and the overall prevalence of hypertension in adults is around 30–45% with 10 million deaths and above 200 million disability-adjusted life years (DALYs), in 2015 [6, 7]. This high prevalence of raised blood pressure is consistent across the world, irrespective of socio-economic status. However, nearly four-fifths of deaths due to CVDs occur in low-and middle-income countries (LMICs) [8] and hypertension disproportionately impacts LMICs [3]. Diseases related to elevated blood pressure have a greater impact on healthcare expenditure, and raised blood pressure and its complications cost an estimated 10% of healthcare spending [9].

Recent studies conducted in Ethiopia show that the prevalence of risk factors of CVDs is increasing rapidly [10]. The prevalence of raised blood pressure in Ethiopia ranges from 19.6% to 30.2% [11, 12]. In general, in lower-and middle-income countries including Ethiopia, the burden of diseases has been shifting to non-communicable diseases from communicable diseases due to unplanned rapid urbanization of rural communities and life style changes in populations in terms of nutrition, physical inactivity, and increase in behaviors such as harmful use of alcohol, tobacco use and drug use in both rural and urban residents. Therefore, prevention, early detection and control programs for hypertension are very important to reduce its impact on health and economy [13].

Besides, prevention and control of hypertension in Ethiopia has not received due attention due to limited health care budget. The health care system gives high priority for maternal health and prevention and control of communicable diseases like tuberculosis, HIV/AIDS, and malaria [14]. Moreover, while civil servants are likely to be at an increased risk for hypertension due to sedentary lifestyle, stress and other factors that are peculiar to civil servants, there is limited information on prevalence and risk factors of hypertension among civil servants in Ethiopia, so much so in the Sidama Zone of southern Ethiopia. Therefore, this study was conducted to determine prevalence and risk factors of hypertension among civil servants in different departments of Sidama Zone, southern Ethiopia.

## Methods and materials

### Study setting

The study was conducted in Sidama zone, south Ethiopia. Sidama zone is one of the eighteen zones and three special districts in the Southern Nations Nationalities and Peoples Region and located 275 km south of Addis Ababa. It consists of 30 districts and 6 town administrations

with a total of 576 kebeles (smallest and lowest administrative units in Ethiopia). Based on the Ethiopian Population Census Report 2007 projections, the total population of the Zone in 2019 has reached 3,893,816.

The Zone administration has a total of 4806 health professionals of different disciplines and 524 health posts, 125 health centers, one general hospital and 13 primary hospitals owned by the government. There are also 21 private medium and three nongovernmental (NGO) clinics, and 63 private drug stores. The overall potential health service coverage of the zone by public health facilities is 90.3%. Hawassa city is the capital city of the Sidama Zone Administration. Currently there are 26 zonal departments/ offices administered under Sidama Zone Administration and all are located in the capital, Hawassa city. Each department/ office has its own distinct responsibilities for serving the community.

## Study design and population

An institution-based cross-sectional study was conducted from March 1–30, 2019 among civil servants of Sidama Zone. The source population for the study comprised of all civil servants who were working in different departments of Sidama Zone Administration while the study population constituted randomly selected civil servants. Pregnant civil servants were excluded from the study, as they might have preeclampsia or pregnancy induced hypertension.

## Sample size determination

The sample size was calculated using Epi-Info™ version 7. The sample size needed to estimate the prevalence of hypertension was calculated assuming the anticipated prevalence of hypertension to be 37.7% based on a study among workers in Akaki Steel Factory, Addis Ababa [15], margin of error (d) of 0.05, z-value of 1.96 for a 95% confidence level and a 10% non-response rate. Accordingly, the calculated sample size was 402. The sample size needed to identify the risk factors associated with hypertension was calculated considering variables significantly associated with hypertension in previous studies [15–17] and fixing the level of confidence at 95%, power at 80%, ratio of unexposed-to-exposed at 1 and anticipated nonresponse rate at 10%. In the study among workers in Akaki Steel Factory, Addis Ababa [15], the prevalence of hypertension among persons who were not current alcohol users (unexposed) was 39.0% and current alcohol use was associated with hypertension with an AOR of 2.10. In the study in northwest Ethiopia [16], the prevalence of hypertension among persons who were not current alcohol users (unexposed) was 28.8% and current alcohol use was associated with hypertension with an AOR of 1.71. Further, in a study among workers of federal ministries of Ethiopia [17], the prevalence of hypertension among persons who were cigarette smokers was 21.5% and cigarette smoking was associated with hypertension with an AOR of 2.34. The sample sizes accordingly calculated were 283, 575 and 263. Hence the sample size of 575 was used as the final sample size for the present study as it would suffice to address all objectives of the study.

## Sampling procedure

The study participants were selected from the complete list of all civil servants which was obtained from Sidama Zone Finance and Economic Development Department's payroll. The list on the payroll served as a sampling frame.

Sidama Zone Administration comprises 26 departments/ zonal offices with a total of 1,341 civil servants, who were on duty during November 2018 [Personal Communication, Sidama Zone Finance and Economic Development Department, 2018]. All departments were included in the study and the total sample size was apportioned for each department proportional to the

number of employees of the respective departments. For departments which had small number of employees and, as a result, for which the allocated sample sizes were very small, there was over-sampling to ensure a better representation of the workers in such departments. Therefore, all departments with sample size less than 15 were boosted to 15 and departments with sample sizes greater than 35 were downsized to 35. Then, civil servants were stratified into three categories based on their responsibility as higher level or executives, technical workers or experts and supportive staff (drivers, secretaries, guards, janitors, etc.) in order to give sufficient representation for all employees. The sample size allocated to a given department was also proportionally allocated to the three strata within a department. Proportionally allocated study subjects were selected using simple random sampling.

When randomly selected employees were absent during data collection, the interview was conducted during the second or third visits. But, study subjects who were absent in the third visit for different reason or unwilling to participate in the study were considered non-respondents.

## Data collection tools and procedures

Data were collected using a structured, face-to-face interviewer-administered questionnaire and standard physical measurements as per the WHO STEPwise approach to chronic disease risk factor surveillance (STEPS) [18]. The questionnaire covered such issues as sociodemographic characteristics of the respondents; behavioral variables such as substance use, dietary habit and physical activity patterns; personal history of diabetes, and family history of hypertension. The questionnaire was prepared in English language and translated to a local language (Amharic) and translated back to English to check for consistency (See supporting information S1 and S2 Files).

Physical measurements included weight, height, waist circumference, hip circumference and blood pressure measurement but did not include biochemical measurements due to resource constraints. Standard functional mercury sphygmomanometers and stethoscope were used to measure the blood pressure with calibration or adjustment to zero level for each measurement and participant. Before measuring blood pressure, the participant was asked to rest for five to ten minutes in a sitting position with legs uncrossed. They were also asked to confirm that they had not smoked or consumed caffeine-containing products for at least 30 minutes prior to measurements. If participants had smoked or used caffeine within 30 minutes prior to blood pressure measurement, data collectors told them with justification to stay there at least for 30 minutes. Three consecutive measurements were taken in an interval of at least five minutes. Mean systolic and diastolic blood pressures were determined by averaging the second and third measurements [3, 19].

Body weight was measured using standardized adult weighing scale with the subjects standing, arms hanging naturally at the sides, wearing light clothing and without footwear. Similar weighing scale was used for all subjects, calibrated for each individual and the reading was expressed in kilograms (kgs) to the nearest 0.5 kg [20]. Height of the study subjects was measured in meters using a stadiometer, with the subjects standing upright, facing straight forward, barefooted, and measurement was expressed to the nearest centimeter [21].

Six nurses with a bachelor's degree and two health officers were recruited from public health institutions of the study area for data collection and supervision, respectively. Then, data collectors and supervisors were trained for two days on the objectives of the study, data collection tools, interview techniques, and their roles and responsibilities by the principal investigator. At the end of the training, the questionnaire was pre-tested on 5% of a total sample size on civil servants in Hawassa City Administration before the actual data collection.

Regular supervision of data collectors to ensure completeness and accuracy of the data were made by supervisors and the principal investigator.

## Study variables

The outcome variable was hypertension status which was defined as a systolic blood pressure equal to or above 140 mmHg and/or diastolic blood pressure equal to or above 90 mmHg, or any prior diagnosis of hypertension made by a health professional, and taking antihypertensive drugs [3]. The independent variables were sociodemographic variables such as age, sex, marital status, religion, family size, monthly income, occupation, educational status, and residence; behavioral or lifestyle factors like cigarette smoking, alcohol drinking, physical inactivity, unhealthy diets, additional salt use, khat (*Catha edulis*) chewing, and coffee drinking; disease history such as personal history of diabetes mellitus, and family history of hypertension; and metabolic factors like overweight, obesity, and central obesity.

## Definition of variables

***Sedentary lifestyle or physical inactivity*** refers to a type of lifestyle involving little or no physical activity. In the present study, physical inactivity was defined as not meeting WHO recommendations on physical activity for health, i.e., respondents doing less than 150 minutes of moderate-intensity physical activity per week; or <75 minutes of vigorous-intensity physical activity per week; or an equivalent combination of moderate- and vigorous-intensity physical activities per week. ***Moderate-intensity activities*** are activities that require moderate physical effort and cause small increases in breathing or heart rate such as cleaning, gardening and cycling at a regular pace. ***Vigorous-intensity activities*** encompass activities that require hard physical effort and cause large increases in breathing or heart rate like carrying or lifting heavy loads, digging or construction work, sports, fitness or recreational (leisure) activities [18].

***Body Mass Index (BMI)*** is defined as the individual's body mass in kgs divided by the height in meters squared. The present study adopted the WHO classification for categorizing study participants into different weight categories. Accordingly, persons with BMI below 18.5 kg/ $m^2$ were categorized as *underweight*, persons with BMI between 18.5 kg/ $m^2$ and 24.9 kg/ $m^2$ as *normal*, persons with BMI between 25 kg/ $m^2$ and 29.9 kg/ $m^2$ as *overweight*, and persons with BMI greater than or equal to 30 kg/ $m^2$ as *obese* [18].

***Central obesity*** was defined based on waist-to-hip ratio (WHR). Existing recommendations for WHR cut-off to define central obesity among males range from 0.9 to 1.0. For the present study we used a WHR of greater than 0.95 to define central obesity among males. For females, a WHR greater than 0.85 was used to define central obesity. [21–23].

***Current alcohol users*** are participants who reported having consumed any type of alcoholic drink in the past 30 days prior to the survey. Participants who reported binge drinking within the past 30 days, which is on average equivalent to more than 6 standard drinks of alcohol in one sitting for men, or an equivalent of more than 4 standard drinks of alcohol per occasion for women, were categorized as ***heavy alcohol users***. While ***medium alcohol users*** refers to participants who reported alcohol consumption of 4 to 6 standard drinks on average per occasion among men, or 2 to 4 standard drinks per occasion among women, ***low alcohol users*** are participants reporting consumption of less than 4 standard drinks among men, or less than 2 standard drinks of alcohol per occasion among women [18].

***Current smoking*** refers to smoking tobacco products, which are manufactured or locally produced within the last 30 days. Light smoker is a person smoking less than 10 cigarettes in a day, whereas moderate smoker is a person smoking 10 or more but less than 20 cigarettes per day and, heavy smoker is a person smoking greater than 20 cigarettes per day.

*Current khat chewers* are participants who reported having consumed khat in the past 30 days prior to the survey.

## Data analysis procedure

The data were entered into EpiData 3.1 and exported to the Statistical Package for Social Sciences (SPSS) version 20.0 for analysis. All required variable recoding and computations were done prior to the main analysis. Five monthly income categories (income quintiles) were created by dividing the monthly income data into five ordinal income groups. The income quintile served as a measure of the socioeconomic status of the study participants. Descriptive analyses were conducted to obtain descriptive measures for the sociodemographic characteristics and other variables. Binary logistic regression was used to identify risk factors associated with hypertension. The binary logistic regression analysis started with unadjusted analysis in which each potential risk factor was assessed separately for its association with hypertension. Variables with p-values < 0.25 on the unadjusted analysis were entered into a multivariable binary logistic regression model to find out risk factors independently associated with hypertension adjusting for other factors in the model. Effect modification was examined by entering interaction terms into the multivariable model one at a time. We entered in the model interaction terms for BMI and age category, BMI and sex, central obesity and sex, and khat chewing and alcohol use to see if age category modifies the effect of BMI, if sex modifies the effects of BMI and central obesity, and if alcohol use modifies the effect of khat chewing. None of the interaction terms was statistically significant implying absence of a significant effect modification. Multicollinearity between the independent variables was also assessed using multiple linear regression. No evidence of multicollinearity was found as the variance inflation factor (VIF) for all variables was less than 10 and the tolerance statistic was greater than 0.1. The goodness- of- fit of the logistic regression model was assessed using the Hosmer-Lemeshow test, the classification accuracy of the model and pseudo-$R^2$. The presence and strength of association between hypertension and the risk factors was evaluated using adjusted odds ratios (AORs) with 95% CIs. Statistically significant association was declared when the 95% CI of the AOR did not embrace 1.

## Ethics statement

Ethical clearance was obtained from the Institutional Review Board (IRB) at the College of Medicine and Health Sciences of Hawassa University before commencing data collection (Ref. No: IRB/041/11). Informed written consent was obtained from each study participant after explaining the objectives, risks/benefits, rights, confidentiality, nature of the study and the scope of their involvement in the study. Study participants with high blood pressure and not already on follow-up were referred to nearby health facilities for further diagnosis and treatment.

# Results

## Socio-demographic characteristics of the study participants

Out of 575 eligible study participants, a total of 546 civil servants took part in the study resulting in a response rate of 94.9%. Thirteen (2.2%) study participants were not around because of annual leave and 16 (2.8%) were not willing to participate. Among the total respondents, 356 (65.2%) were males and the mean (±standard deviation [SD]) of the age of participants was 37.86 (±9.50) years. See details of the socio-demographic characteristics of the study participants in Table 1.

**Table 1. Sociodemographic characteristics of the study participants in Sidama Zone, south Ethiopia, 2019.**

| Variable (n = 546) | Categories | Number | Percent |
|---|---|---|---|
| Sex | Male | 356 | 65.2 |
| | Female | 190 | 34.8 |
| Age | 18–29 | 119 | 21.8 |
| | 30–39 | 194 | 35.5 |
| | 40–49 | 149 | 27.3 |
| | ≥50 | 84 | 15.4 |
| Marital status | Single | 77 | 14.1 |
| | Married | 410 | 75.1 |
| | Divorced | 32 | 5.9 |
| | Widowed | 27 | 4.9 |
| Educational status | Primary education (Grade 1–8) | 8 | 1.5 |
| | Secondary education (Grade 9–12) | 45 | 8.2 |
| | Above secondary education | 493 | 90.3 |
| Task level | High level managerial work | 125 | 22.9 |
| | Experts | 318 | 58.2 |
| | Supportive staff | 103 | 18.9 |
| Family size | 1–5 | 304 | 55.7 |
| | >5 | 242 | 44.3 |
| Family monthly income quintile (birr) | Lowest | 98 | 17.9 |
| | Second lowest | 102 | 18.7 |
| | Middle | 148 | 27.1 |
| | Second highest | 97 | 17.8 |
| | Highest | 97 | 17.8 |

### History of substance use

From the 546 study participants, 76 (13.9%) had smoked cigarettes at least once in their life-time, and only two (0.4%) were currently smoking (in the past four weeks). The two currently smoking participants were smoking daily for more than 20 years as light smokers. Two hundred nineteen (40.1%) of the respondents were past drinkers, while 90 (16.5%) were still drinking.

Concerning khat chewing, 138 (25.3%) participants reported chewing khat at least once in their life time, whereas 69 (12.6%) were current chewers. Further, 515 (94.3%) study participants were coffee drinkers, of whom 479 (93%) were drinking coffee on a daily basis (Table 2).

### Physical activity of the study participants

Among the study participants, 173 (31.7%) reported that they were involved in work related vigorous activities, of whom 94 (54.3%) performed such activities for 4–7 days per week. The rest performed 1–3 days per week. The majority (78%) of the participants had one hour of work-related vigorous activity per day. About 90% of all respondents reported that they practised brisk walking for at least 10 minutes per day for 1–7 days in a week, and 55.8% practised it on daily basis. One hundred and sixteen (21.2%) of the study participants were practising sports and recreational vigorous physical activities, of whom 43% practised such activities for 1–2 days per week. Overall, 385 (70.5%) of the study participants were physically active in one or more categories of physical activities (Table 3).

**Table 2. History of substance use of the study participants in Sidama Zone, south Ethiopia, 2019.**

| Variable (n = 546) | Categories | Number | Percent |
|---|---|---|---|
| Ever smoked cigarette | Yes | 76 | 13.9 |
| | No | 470 | 86.1 |
| Currently smoking cigarettes | Yes | 2 | 0.4 |
| | No | 544 | 99.6 |
| Ever drunk alcohol | Yes | 219 | 40.1 |
| | No | 327 | 59.9 |
| Currently drinking alcohol | Yes | 90 | 16.5 |
| | No | 456 | 83.5 |
| Level of current standard alcohol drink | Not currently drinking | 456 | 83.5 |
| | Low | 51 | 9.3 |
| | Moderate | 39 | 7.1 |
| | Heavy | 0 | 0 |
| Ever chewed khat | Yes | 138 | 25.3 |
| | No | 408 | 74.7 |
| Currently chewing khat | Yes | 69 | 12.6 |
| | No | 447 | 87.4 |
| Khat chewing days per week (n = 69) | 1–2 days | 37 | 53.6 |
| | 3–4 days | 28 | 40.6 |
| | 5 days or more | 4 | 5.8 |
| Coffee drinking | Yes | 515 | 94.3 |
| | No | 31 | 5.7 |
| Days of coffee drinking per week (n = 515) | Daily | 479 | 93.0 |
| | 5–6 days | 7 | 1.4 |
| | 3–4 | 9 | 1.7 |
| | 1–2 | 20 | 3.9 |
| Cups of coffee consumed per day (n = 515) | 1 cup | 86 | 16.7 |
| | 2 cups | 149 | 28.9 |
| | 3 and more cups | 280 | 54.4 |

## Dietary practice of the study participants

Among the participants 511 (93.6%) had the habit of consuming fruits and 538 (98.5%) had the habit of vegetable consumption; however, only 45 (8.8%) were eating fruits and 176 (32.7%) were eating vegetables on daily basis. Eighty six (15.8%) were using fatty foods of animal origin and 50 (9.2%) were using additional salt in their diet (Table 4).

## Anthropometric characteristics of the study participants

Regarding the body mass index (BMI), 233 (42.7%) of the study participants were overweight and 97 (17.8%) were obese. Concerning the central obesity, 275 (50.4%) of the screened participants had high waist-to-hip ratio (WHR > 0.95 for males and > 0.85 for females). The mean (±SD) of BMI of the study participants was 26.2 (± 4.0) and the mean (±SD) of WHR was 0.91 (± 0.08) (Table 5).

## Prevalence of hypertension

The overall prevalence of hypertension among civil servants in all departments of Sidama Zone Administration was 24.5% (95% CI: 23.3% - 25.6%), whereas the prevalence of newly screened hypertension was 15.2% (95% CI: 12.4% - 18.4%). The mean (±SD) systolic blood

**Table 3. Physical activities of the study participants in Sidama Zone, south Ethiopia, 2019.**

| Variable (n = 546) | Categories | Frequency | Percent |
|---|---|---|---|
| Work related vigorous activity | Yes | 173 | 31.7 |
| | No | 373 | 68.3 |
| Vigorous work related activity days per week (n = 173) | 1–3 days | 79 | 45.7 |
| | 4–7 days | 94 | 54.3 |
| Vigorous work related activity hours per day (n = 173) | One hour | 135 | 78.0 |
| | Two hours | 28 | 16.0 |
| | 3–4 hours | 10 | 6.0 |
| Brisk walking for at least for 10 minutes per day | Yes | 496 | 90.8 |
| | No | 50 | 9.2 |
| Frequency of brisk walking days per week (n = 496) | Daily | 277 | 55.8 |
| | 5–6 days | 82 | 16.5 |
| | 3–4 days | 90 | 18.2 |
| | 1–2 days | 47 | 9.5 |
| Vigorous-intensity sports or recreational activity | Yes | 116 | 21.2 |
| | No | 430 | 78.8 |
| Frequency of vigorous intensity sport activity per week (n = 116) | Daily | 9 | 7.8 |
| | 5–6 days | 12 | 10.3 |
| | 3–4 days | 45 | 38.8 |
| | 1–2 days | 50 | 43.1 |
| Time spent for vigorous sport activity in minutes per day (n = 116) | 10–30 | 62 | 53.4 |
| | >30 | 54 | 46.6 |
| Means of transportation | Vehicle with engine | 525 | 96.2 |
| | On foot | 17 | 3.1 |
| | Bicycle | 4 | 0.1 |
| Physical activity status | Active | 385 | 70.5 |
| | Inactive | 161 | 29.5 |

pressure was 120.22 (± 15.03) mmHg and the mean (±SD) diastolic blood pressure was 80.04 ± 9.51 mmHg. The majority (76.1%) of hypertension cases were males.

Four hundred twenty nine (78.6%) of the participants had ever been measured their blood pressure. Among the 134 hypertensive cases, 51 (38%) were aware of being hypertensive prior

**Table 4. Dietary practice of the study participants in Sidama Zone, south Ethiopia, 2019.**

| Variable (n = 546) | Categories | Frequency | Percent |
|---|---|---|---|
| Fruit consumption | Yes | 511 | 93.6 |
| | No | 35 | 6.4 |
| Fruit consumption days per week (n = 511) | ≤ 6 days | 466 | 91.2 |
| | Daily | 45 | 8.8 |
| Vegetable consumption | Yes | 538 | 98.5 |
| | No | 8 | 1.5 |
| Vegetable consumption days per week (n = 538) | ≤ 6 days | 362 | 67.3 |
| | Daily | 176 | 32.7 |
| Use of fatty food of animal origin | Yes | 86 | 15.8 |
| | No | 460 | 84.2 |
| Using additional salt on foods | Yes | 50 | 9.2 |
| | No | 496 | 90.8 |

**Table 5. Anthropometric characteristics of the study participants in Sidama Zone, south Ethiopia, 2019.**

| Variable (n = 546) | Categories | Frequency | Percent |
|---|---|---|---|
| BMI category | Underweight | 12 | 2.2 |
| | Normal | 204 | 37.4 |
| | Overweight | 233 | 42.7 |
| | Obese | 97 | 17.8 |
| Mean (± SD) BMI | | 26.2 (± 4.0) | |
| WHR | High | 275 | 50.4 |
| | Low | 271 | 49.6 |
| Mean (± SD) WHR | | 0.91 (± 0.08) | |

BMI, body mass index; SD, standard deviation; WHR, waist-to-hip ratio

to the study, of whom 24 (47%) were using anti-hypertensive medication; however, only 8 (33.3%) of those on medication had got their blood pressure controlled (Table 6).

## Risk factors for hypertension among civil servants

Age, sex, educational status, income quintile, marital status, family size, task level, harmful use of alcohol, tobacco use, physical activity status, means of transportation, use of fatty food of animal origin, fruit and vegetable consumption, use of additional salt in food, khat use, coffee drinking, having diabetes mellitus, family history of hypertension, waist-to-hip ratio, BMI and

**Table 6. Prevalence of hypertension among the study participants in Sidama Zone, south Ethiopia, 2019.**

| Variable (n = 546) | Categories | Frequency | Percent |
|---|---|---|---|
| Hypertension | Yes | 134 | 24.5 |
| | No | 412 | 75.5 |
| Ever been measured blood pressure prior to screening | Yes | 429 | 78.6 |
| | No | 117 | 21.4 |
| Known hypertensive cases (n = 134) | Yes | 51 | 38.0 |
| | No | 83 | 62.0 |
| Hypertension cases on medication (n = 51) | Yes | 24 | 47.0 |
| | No | 27 | 53.0 |
| Newly screened hypertension case | Yes | 83 | 15.2 |
| | No | 463 | 84.8 |
| Hypertension by both systolic and diastolic (n = 134) | Yes | 61 | 45.5 |
| | No | 73 | 54.4 |
| Family history of hypertension | Yes | 191 | 35.0 |
| | No | 355 | 65.0 |
| Controlled hypertension (n = 24) | Yes | 8 | 33.3 |
| | No | 16 | 66.7 |
| Personal history of diabetes mellitus | Yes | 31 | 5.7 |
| | No | 515 | 94.3 |
| Level of blood pressure (n = 522)* | Normal | 322 | 61.7 |
| | Pre-hypertension | 90 | 17.2 |
| | Stage I | 79 | 15.1 |
| | Stage II | 31 | 5.9 |

*The 24 hypertensive cases who were on medication were excluded from the analysis as the medication would affect the correct classification level of blood pressure.

level of knowledge were considered in the crude (bi-variable) analyses. As revealed by the bi-variable analysis, age, sex, marital status, family size, income quintile, harmful use of alcohol, khat use, use of fatty food of animal origin, using additional salt in meals, having diabetes mellitus, BMI and central obesity were found to be candidate variables for the multivariable model (p<0.25). In the multivariable model, male sex, old age, current moderate alcohol drinking, current khat chewing, being obese and central obesity were significantly associated with hypertension (p<0.05).

The odds of hypertension was 4.31 times higher in males as compared to female study participants (AOR: 4.31; 95% CI: 1.84–10.09). The odds of hypertension increased 2.97 times for study participants who had a habit of current khat chewing (AOR: 2.97, 95% CI: 1.38–6.40) than study participants who were not current khat chewers. Besides, being in old age (AOR: 4.41; 95% CI: 1.19–16.26), having moderate level of current alcohol consumption (AOR: 4.85; 95% CI: 1.73–13.61) and being obese (AOR: 5.94; 95% CI: 1.26–27.86) were found to increase the odds of hypertension among civil servants. Further, civil servant with central obesity had 3.57 times higher odds of hypertension relative to those having no central obesity (AOR: 3.57; 95% CI: 1.80–7.07) (Table 7).

## Discussion

This study was conducted to determine prevalence and risk factors of hypertension among civil servants in different departments of Sidama Zone, southern Ethiopia. The prevalence of hypertension among the civil servants in the zonal administration was 24.5%. Old age, male sex, current khat chewing, current moderate alcohol drinking, obesity and central obesity were found to be the risk factors of hypertension.

The overall prevalence of hypertension was 24.5% which is consistent with other studies conducted in Bahir Dar city (25.1%) and in Addis Ababa (25%) [24, 25]. However, this finding is lower than that of the studies in Gondar city (28.3%) and Jigjiga town (28.3%) [26, 27]. The reason for these discrepancies might be that all study participants in Gondar city were 35 years old and above, and the study participants in Jigjiga town were 25 to 65 years old, whereas the participants in the present study were 18 years and above. Different studies indicate that the prevalence of hypertension increases in older age [16, 17, 24, 26, 28–32]. The other potential reason for these differences might be the study settings. Both previous studies were community based, whereas our study is institution based.

However, the prevalence of hypertension in this study is higher than that of community-based studies conducted in Bedele town (16.9%) and Durame town (22.4%) [32, 33]. Similarly, a hospital-based study conducted in Jimma showed lower prevalence (13.2%) than the current study [34].

These variations might be due to the fact that the first two studies were community-based in which the study participants were with different types of occupations including farmers and daily laborers, whereas in the present study the participants were civil servants. The participants of the study in Jimma were patients who came from rural and urban areas but the participants in the present study were from urban setting which might explain the discrepancy, at least partly. The population in our study was civil servants with more sedentary life style related to their work condition, which puts the population addressed by the present study at a higher risk than that of the population in previous studies. The magnitude of hypertension is higher in urban than in rural settings mainly because of contextual and behavioral factors associated with urban environments such as dietary changes and sedentary lifestyle that together form a complex system conducive for developing hypertension [35].

Table 7. Risk factors of hypertension among civil servants in Sidama Zone, south Ethiopia, 2019.

| Variable (n = 546) | Hypertension status | | COR of 95% CI | AOR of 95% CI |
|---|---|---|---|---|
| | Yes (%) | No (%) | | |
| Sex | | | | |
| Male | 102 (28.7) | 254 (71.3) | 1.98(1.27, 3.09) | 4.31(1.84, 10.09)** |
| Female | 32 (16.8) | 158 (83.2) | 1 | 1 |
| Age in years | | | | |
| 18–29 | 10 (8.4) | 109 (91.6) | 1 | 1 |
| 30–39 | 36 (18.6) | 158 (81.4) | 2.48(1.18, 5.21) | 1.96(0.65, 5.87) |
| 40–49 | 52 (34.9) | 97 (65.1) | 5.84(2.81, 12.12) | 2.14(0.65, 7.03) |
| >50 | 48 (42.9) | 36 (57.1) | 8.17(3.75, 17.80) | 4.41(1.19, 16.26)* |
| Marital status | | | | |
| Single | 10 (13.0) | 67 (87.0) | 1 | 1 |
| Married | 109 (26.6) | 301 (73.4) | 2.42(1.20, 4.88) | 1.15(0.37, 3.55) |
| Divorced | 8 (25.0) | 24 (75.0) | 2.23(0.78, 6.31) | 1.24(0.25, 5.96) |
| Widowed | 7 (25.9) | 20 (74.1) | 2.34(0.79, 6.95) | 2.40(0.42, 13.61) |
| Family size | | | | |
| 1–5 | 60 (19.7) | 244 (80.3) | 1 | 1 |
| >5 | 74 (30.6) | 168 (69.4) | 1.79(1.20, 2.54) | 0.60(0.29, 1.25) |
| Income quintile (n = 542) | | | | |
| Lowest | 17 (17.3) | 81 (82.7) | 1 | 1 |
| Second lowest | 25 (24.5) | 77 (75.5) | 1.54(0.77, 3.08) | 0.54(0.19, 1.49) |
| Middle | 45 (30.4) | 103 (69.6) | 2.08(1.10, 3.90) | 0.87(0.34, 2.24) |
| Second highest | 25 (25.8) | 72 (74.2) | 1.65(0.82, 3.30) | 0.92(0.30, 2.75) |
| Highest | 22 (22.7) | 75 (77.3) | 1.39(0.69, 2.83) | 0.60(0.20, 1.76) |
| Current level of alcohol use | | | | |
| No | 93 (20.4) | 363 (79.6) | 1 | 1 |
| Low | 19 (37.3) | 32 (62.7) | 2.31(1.25, 4.27) | 1.57(0.60, 4.15) |
| Moderate | 22 (56.4) | 17 (43.6) | 5.05(2.57, 9.89) | 4.85(1.73, 13.61)** |
| Current khat Chewing | | | | |
| Yes | 71 (50.7) | 67 (49.3) | 3.93(2.33, 6.61) | 2.97(1.38, 6.40)** |
| No | 63 (20.8) | 345 (79.2) | 1 | 1 |
| Eating fatty food of animal origin | | | | |
| Yes | 28 (32.6) | 58 (67.4) | 1.61(0.97, 2.65) | 2.12(0.95, 4.71) |
| No | 106 (23.0) | 354 (77.0) | 1 | 1 |
| Use of additional salt on foods | | | | |
| Yes | 8 (16.0) | 42 (84.0) | 1.78(0.81, 3.91) | 0.76(0.25, 2.28) |
| No | 126 (25.4) | 370 (74.6) | 1 | 1 |
| Having diabetes mellitus | | | | |
| Yes | 15 (48.4) | 16 (51.6) | 3.12(1.49,6.49) | 1.42(0.44, 4.58) |
| N o | 119 (23.1) | 396 (76.9) | 1 | 1 |
| BMI | | | | |
| <18.5 | 5 (41.7) | 7 (58.3) | 1 | 1 |
| 18.5–24.99 | 17 (8.3) | 187 (91.7) | 1.56(0.19, 12.62) | 0.07(0.01, 0.31) |
| 25–29.99 | 30 (12.9) | 203 (87.1) | 2.94(0.37, 13.42) | 0.08(0.01, 0.32) |
| >30 | 82 (84.5) | 15 (15.5) | 12.15(3.52, 27.1) | 5.94(1.26, 27.86)** |
| Central obesity | | | | |
| Yes | 100 (36.4) | 175 (63.6) | 3.98(2.57, 6.15) | 3.57(1.80, 7.01)** |

(*Continued*)

**Table 7.** (*Continued*)

| Variable (n = 546) | Hypertension status | | COR of 95% CI | AOR of 95% CI |
|---|---|---|---|---|
| | Yes (%) | No (%) | | |
| No | 34 (12.5) | 237 (87.5) | 1 | 1 |

1 indicates the reference categories; a single asterisk (*) indicates a significant association (p-value < 0.05); double asterisk (**) indicates a highly significant association (p-value <0.01).

BMI, body mass index.

Hosmer and Lemeshow test: chi-square = 20.305, degree of freedom = 8, p-value = 0.009; Nagelkerke $R^2$ = 0.618; overall classification accuracy = 89.3%.

Different studies have showed that several factors have significant association with hypertension. In the present study, the odds of hypertension in males was more than four times higher than that in the females. This is similar with findings from several previous studies conducted in different areas. The study conducted in Tigray among civil servants showed the odds of hypertension in males to be two times higher than that of females [31], which was similar to the studies among students of Gondar University [36] and civil servants in Addis Ababa [30]. Moreover, studies conducted in China, India and Nepal revealed similar findings [37–39]. The reason for these differences might be due to the exposure towards different behavioral risk factors for hypertension, which is higher among males than among females in most areas. Moreover, the molecular mechanisms underlying vasculature, nervous system, and kidney functions, which lead to hypertension and the pathways for the control of blood pressure may explain the differences between the sexes [40]. In contrast, some studies from Sub-Saharan Africa report the prevalence of hypertension to be higher in females than in males [41].

The current study also revealed that old age was significantly associated with hypertension. Similar findings were reported from the study conducted at Hawassa University [29], in Tigray [31], Addis Ababa [17], Bahir Dar city [24], Durame town [32], Gondar city [26], Jmma town [42], and Bedele town of Ethiopia [33]. Similarly, studies conducted in the United States and China reported results similar to the current study [28, 37]. This might be due to the physiological change of blood vessels as the individual's age is increased, in which blood vessels might lose flexibility due to hardening of the arteries as age advances [43].

Current alcohol drinking was significantly associated with hypertension. Studies conducted in Addis Ababa [17], Gondar city [26], and Jimma town [42] showed similar significant associations. Likewise, studies conducted in Ghana [44], Nepal [39], and India [38] also showed significant association between alcohol drinking and hypertension. This might be due to the fact that alcohol consumption raises the amount of lipids in the bloodstream, which can damage the arteries. This in turn leads to hardening of arteries that increases the risk of blood clots which can raise blood pressure. Another reason might be that alcohol has high calories and sugar, which can increase the risk of high blood pressure in the long term by adding to the body fat [45].

In this study, BMI $\geq$ 30 Kg/m$^2$ (obesity) was significantly associated with hypertension. Similar findings were reported from studies in Hawassa University [29], Tigray region [31] and Addis Ababa [17]. Studies conducted in Indian and Nepal also reported consistent findings [38, 39]. The reason for this might be related to the increased oxygen and nutrient demand caused by the additional fat or adipose tissue in the body, which requires the blood vessels to circulate more blood to the extra fat tissue. As body weight increases there is also an increase in blood circulation to different vital body parts "due to increased metabolic rate and growth of the organs and tissues in response to their increased metabolic demands," which in turn increases pressure on the walls of the arteries [43].

The result of this study also indicated that central obesity had an association with hypertension. This finding agreed with studies conducted in Bedele town [33], Addis Ababa [12] and India [38]. Different mechanisms that can explain the association between central obesity and hypertension have been proposed. The two major mechanisms appear to be hyperactivity of the sympathetic nervous system (SNS) and activation of the renin-angiotensin system (RAS) [46, 47]. When fat is accumulated in the body, the adipokine leptin is thought to cause hyperactivation of the SNS [46, 47]. Long-term hyperactivity of the SNS may cause hypertension through peripheral vasoconstriction and increased absorption of sodium by the kidneys [47]. On the other hand, there is evidence showing that intra-abdominal fat increases the production of angiotensinogen, one of the proteins of the RAS. This is thought to be one of the mechanisms by which central obesity leads to hypertension [46, 47].

Moreover, this study showed that current khat chewing was significantly associated with hypertension. Studies conducted in Jimma and Gondar also reported a significant association between khat chewing and hypertension [16, 48]. This is thought to be due to cathinone (a stimulant chemical in khat) which increases blood pressure through noradrenaline release, similar to amphetamine, to produce vasoconstriction [49].

The present study has some limitations. First, the present study was conducted among civil servants working at a zonal level and in an urban setting. Hence, the results may not be generalizable to civil servants working at lower (district and kebele) level and in rural settings. Second, the results might have been affected by reporting bias. It is possible to have deliberate misreporting of lifestyle related factors such as alcohol drinking, khat chewing and smoking cigarette (social desirability bias). Hence, the magnitude of these risk factors might have been underestimated and as such the association of these factors with hypertension might have been attenuated. Further, the relative concentration of responses in the middle income quintile might indicate misreporting of income. That might have led to the absence of association between income quintile and hypertension.

In conclusion, one in four civil servants in our study area are hypertensive. Males, older individuals, current moderate alcohol consumers, current khat chewers, and individuals with obesity and central obesity have higher odds of hypertension. Prevention and control of hypertension shall involve promotion of healthy lifestyles such as weight management, regular physical activity and quitting or cutting down on harmful use of substances such as alcohol and khat.

## Supporting information

**S1 File.**
(PDF)

**S2 File.**
(PDF)

**S1 Dataset.**
(SAV)

## Acknowledgments

The authors thank the School of Public Health of Hawassa University for providing oversight for the conduct of the study. The study participants also deserve thanks for willfully taking part in the study. Finally, the authors are grateful to Sidama Zone Health Department for providing support during data collection.

## Author Contributions

**Conceptualization:** Bedilu Badego, Ayalew Astatkie.

**Data curation:** Bedilu Badego, Amanuel Yoseph, Ayalew Astatkie.

**Formal analysis:** Bedilu Badego, Amanuel Yoseph, Ayalew Astatkie.

**Investigation:** Bedilu Badego, Ayalew Astatkie.

**Methodology:** Bedilu Badego, Amanuel Yoseph, Ayalew Astatkie.

**Project administration:** Bedilu Badego.

**Resources:** Bedilu Badego.

**Software:** Bedilu Badego, Amanuel Yoseph, Ayalew Astatkie.

**Supervision:** Bedilu Badego, Ayalew Astatkie.

**Validation:** Bedilu Badego, Ayalew Astatkie.

**Visualization:** Bedilu Badego, Amanuel Yoseph, Ayalew Astatkie.

**Writing – original draft:** Bedilu Badego, Amanuel Yoseph.

**Writing – review & editing:** Ayalew Astatkie.

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
