## [Decision Letter · Decision Letter 0]

10 Feb 2020

PONE-D-19-34743

Prevalence and risk factors of hypertension among civil servants in Sidama Zone, south Ethiopia

PLOS ONE

Dear Mr. Badego,

Thank you for submitting your manuscript to PLOS ONE. After careful consideration, we feel that it has merit but does not fully meet PLOS ONE’s publication criteria as it currently stands. Therefore, we invite you to submit a revised version of the manuscript that addresses the points raised during the review process.

In addition to all the issues raised by the two reviewers and the comments on the attached pdf file, please ensure that you address the following:

1. There are many grammatical errors. These MUST all be corrected before re-submitting the manuscript. Otherwise the manuscript will not be fit for publication in PLOSONE.

2. There are a number of additional statistical analyses that must be completed and results appropriately presented

3. The research data should be submitted with the revised copy of the manuscript.

4. Please follow the author guidelines in writing the declaration section of the manuscript.

5. The logistic regression needs to be described in more detail and the results of the Odds rations appriapriately interpreted.

We would appreciate receiving your revised manuscript by Mar 26 2020 11:59PM. To enhance the reproducibility of your results, we recommend that if applicable you deposit your laboratory protocols in protocols.io, where a protocol can be assigned its own identifier (DOI) such that it can be cited independently in the future. For instructions see: http://journals.plos.org/plosone/s/submission-guidelines#loc-laboratory-protocols

We look forward to receiving your revised manuscript.

Kind regards,

Agricola Odoi, BVM, MSc, PhD, FAHA, FACE

Academic Editor

PLOS ONE

Journal Requirements:

3. Please provide additional details regarding participant consent. In the ethics statement in the Methods and online submission information, please ensure that you have specified (1) whether consent was informed and (2) what type you obtained (for instance, written or verbal). If your study included minors, state whether you obtained consent from parents or guardians. If the need for consent was waived by the ethics committee, please include this information.

4. Please include additional information regarding the survey or questionnaire used in the study and ensure that you have provided sufficient details that others could replicate the analyses. For instance, if you developed a questionnaire as part of this study and it is not under a copyright more restrictive than CC-BY, please include a copy, in both the original language and English, as Supporting Information.

5. In your Methods section, please provide additional information on how monthly income was categorised.

6. We noticed you have some minor occurrence of overlapping text with the following previous publication(s), which needs to be addressed:

- Hall, John E. Guyton and Hall textbook of medical physiology e-Book. Elsevier Health Sciences, 2015.

The text that needs to be addressed involves some sentences of the Discussion.

In your revision ensure you cite all your sources (including your own works), and quote or rephrase any duplicated text outside the methods section. Further consideration is dependent on these concerns being addressed.

7. We note that you have stated that you will provide repository information for your data at acceptance. Should your manuscript be accepted for publication, we will hold it until you provide the relevant accession numbers or DOIs necessary to access your data. If you wish to make changes to your Data Availability statement, please describe these changes in your cover letter and we will update your Data Availability statement to reflect the information you provide.

Reviewers' comments:

Reviewer's Responses to Questions

**Comments to the Author**

1. Is the manuscript technically sound, and do the data support the conclusions?

Reviewer #1: Yes

Reviewer #2: Yes

2. Has the statistical analysis been performed appropriately and rigorously? 

Reviewer #1: Yes

Reviewer #2: Yes

3. Have the authors made all data underlying the findings in their manuscript fully available?

Reviewer #1: Yes

Reviewer #2: Yes

4. Is the manuscript presented in an intelligible fashion and written in standard English?

Reviewer #1: Yes

Reviewer #2: Yes

5. Review Comments to the Author

Reviewer #1: The abstract needs some more transition. The authors start with hypertension being the leading cause of death and disability, then the next sentence notes two thirds (note missing the 's' at the end in the text) of those with it live in low and middle income countries, then say civil servants are understudied. That last thought about understudied civil servants does not follow logically. It is more the case that it is a relatively convenient population for the purposes of this study. It is not really representative of low and middle income country population, but it is feasible to study them.

Worth noting that if it is the case that 67% of cases are in low and middle income countries (as per World Bank definitions) that means they are actually under the global average as (last I checked) low and middle income country population was about 85% of global population. This would make sense given the lower median age in low and middle compare to high income.

Page 11 as read 5 as numbered, control of communicable diseases ... maternal health. That is not a communicable disease and it is implied by the way this is written that it is.

While there are factors that could be related to job related stress being in government service, it is probably less stressful than being a teff farmer in Wollo since at least you know you have a monthly salary!

Page 12 in order, 6 as numbered, you know (and I know) what a kebele is, but for general readership, you should explain where it sits in the administrative structure; for your sampling purposes, it is the lowest level at which public services are delivered. You have multiple villages in a kebele which is something like a township, then kebeles add up to woreda which are like a county, and these aggregate up into zones,....You target the zonal administrative center, but it is delivering down to woreda / kebele / village.

Numbered page 8, we generally don't call it a well structured survey. Just structured (as compared to semi-structured or unstructured).

What do you make of the over-large share of people who are in the middle quintile? It makes me suspect there is some strategic response here. By the nature of the measure, you should be getting 20% across these. Rather than admit to a low salary and feel ashamed or a high salary that might draw attention to oneself which is risky, the safe response is middle. It makes me think of the survey results in the United States where 70% of households consider themselves middle income.

I don't know that it is worth including the smoked once or chewed once measures. I would think that has zero implication for hypertension.

The coffee result is interesting, but how can you be in Ethiopia and not enjoy some coffee!

The definition of vigorous might not be standard across people. Be clearer about what this meant.

Explain the distinction between obesity and central obesity.

Page 23 you have eating animal fat, which is not what you said before.

Page 28, consistent with your results I would think the lack of alcohol in Jigjiga town may have something to do with this result, though the khat use may be higher than where you are studying.

Reviewer #2: In the abstract it states that hypertension is the leading cause of death among adults globally. Cardiovascular disease is the leading cause of death and hypertension is a risk factor for cardiovascular disease. Please consider revising the language in the abstract to reflect the information presented in the introduction.

Hypertension is a public health concern globally and this manuscript adds to the body of literature as it relates to chronic disease. This manuscript is written in a manner that is consistent with the English language.

6. PLOS authors have the option to publish the peer review history of their article (what does this mean?). If published, this will include your full peer review and any attached files.

Reviewer #1: No

Reviewer #2: No

---

## [Author Response · Author response to Decision Letter 0]

3 Apr 2020

Point-by-point responses to reviewers’ and editor’s comments

Reviewer 1

Comment 1: The abstract needs some more transition. The authors start with hypertension being the leading cause of death and disability, then the next sentence notes two thirds (note missing the 's' at the end in the text) of those with it live in low and middle income countries, then say civil servants are understudied. That last thought about understudied civil servants does not follow logically. It is more the case that it is a relatively convenient population for the purposes of this study. It is not really representative of low and middle income country population, but it is feasible to study them.

Worth noting that if it is the case that 67% of cases are in low and middle income countries (as per World Bank definitions) that means they are actually under the global average as (last I checked) low and middle income country population was about 85% of global population. This would make sense given the lower median age in low and middle compare to high income.

Authors’ response: Thank you a lot for these valuable comments. We have accepted the comments and did the required revision. The abstract has been edited to be more logical and more coherent. We also have removed irrelevant text. 

Comment 2: Page 11 as read 5 as numbered, control of communicable diseases ... maternal health. That is not a communicable disease and it is implied by the way this is written that it is.

Authors’ response: Thank you for this important comment. We have revised the description to avoid ambiguity. 

Comment 3: While there are factors that could be related to job related stress being in government service, it is probably less stressful than being a teff farmer in Wollo since at least you know you have a monthly salary!

Authors’ response: Thank you for the genuine and plausible comment. However, we feel that the level of stress in government employees would be higher due to different stressful responsibilities at work place and also due to imbalance between salary and monthly expenditure for living (i.e. house rent, tuition fee for children, food expenditure, transportation cost, etc).

Comment 4: Page 12 in order, 6 as numbered, you know (and I know) what a kebele is, but for general readership, you should explain where it sits in the administrative structure; for your sampling purposes, it is the lowest level at which public services are delivered. You have multiple villages in a kebele which is something like a township, then kebeles add up to woreda which are like a county, and these aggregate up into zones,....You target the zonal administrative center, but it is delivering down to woreda / kebele / village.

Authors’ response: Dear reviewer, thank you for your comment. A kebele is the smallest and lowest administrative unit in Ethiopia. It is part of a woreda (district), itself usually part of a zone, which in turn are grouped into one of the regions of the Federal Democratic Republic of Ethiopia. Dear reviewer, we kindly request you to look at the study area part of this paper. Now we have indicated what a kebele refers to in the study setting sub-section of the methods and materials section.

Comment 5: Numbered page 8, we generally don't call it a well-structured survey. Just structured (as compared to semi-structured or unstructured).

Authors’ response: Thank you for pointing this out. We now have removed the adverb “well” from the description. 

Comment 6: What do you make of the over-large share of people who are in the middle quintile? It makes me suspect there is some strategic response here. By the nature of the measure, you should be getting 20% across these. Rather than admit to a low salary and feel ashamed or a high salary that might draw attention to oneself which is risky, the safe response is middle. It makes me think of the survey results in the United States where 70% of households consider themselves middle income.

Authors’ response: Dear reviewer, thank you for your important comment; we also worried about it before actual data collection. As you correctly pointed out, there is a possibility of reporting bias. Cognizant of this, data collectors were told during the training to properly explain to the study participants the importance of the question about income and to elicit genuine response from the respondents. Yet, that wouldn’t preclude the possibility of bias in reporting income. We now have addressed this issue as a possible limitation of the study in the discussion section (second paragraph from the last). 

Comment 7: I don't know that it is worth including the smoked once or chewed once measures. I would think that has zero implication for hypertension.

Authors’ response: Dear reviewer, thank you for your important comment. We were also cognizant of that fact. In our analysis, we found that 76/546 (13.9%) of the respondents had ever smoked cigarette and 178/546 (25.3%) had ever chewed khat. These results were used only for descriptive purpose of substance use behaviour. We used current cigarette smoking and current khat chewing in the multivariable analysis. Dear reviewer, we kindly request you to look at Table 7 of the paper. 

Comment 8: The coffee result is interesting, but how can you be in Ethiopia and not enjoy some coffee!

Authors’ response: We want to thank the reviewer once again for the genuine comment. It is still possible to find a small fraction of individuals in Ethiopia who do not drink coffee for various reasons. 

Comment 9: The definition of vigorous might not be standard across people. Be clearer about what this meant.

Authors’ response: Thank you for this important comment. We now have included a “Definition of variables” part in the Methods and Materials section of the manuscript and have clarified such variables and terminologies. Accordingly, vigorous-intensity activity has been defined as “activities that require hard physical effort and cause large increases in breathing or heart rate like carrying or lifting heavy loads, digging or construction work, sports, fitness or recreational (leisure) activities”.

Comment 10: Explain the distinction between obesity and central obesity.

Authors’ response: Thank you for this comment too. We have clarified the distinction between the two variables in the “Definition of variables” part.

Comment 11: Page 23 you have eating animal fat, which is not what you said before.

Authors’ response: Dear reviewer thank you for your important comment. Now we have used the term “use of fatty food of animal origin” consistently throughout the manuscript.

Comment 12: Page 28, consistent with your results I would think the lack of alcohol in Jigjiga town may have something to do with this result, though the khat use may be higher than where you are studying.

Authors’ response: Dear reviewer thank you for your important comment. Yes, what you pointed out is a possibility. Another reason might be difference in the population they studied as compared to ours.

Reviewer 2 

Comment 1: In the abstract it states that hypertension is the leading cause of death among adults globally. Cardiovascular disease is the leading cause of death and hypertension is a risk factor for cardiovascular disease. Please consider revising the language in the abstract to reflect the information presented in the introduction.

Authors’ response: Thank you for this comment. This comment was also shared by Reviewer 1 (Comment 1). As per the comment both from Reviewers 1 and 2, the abstract has been edited to be more logical and more coherent. We also have removed irrelevant text.

Comment 2: Hypertension is a public health concern globally and this manuscript adds to the body of literature as it relates to chronic disease. This manuscript is written in a manner that is consistent with the English language. 

Authors’ response: Thank you for your kind remark. 

 

Editor’s comments & journal requirements

Journal requirements

Requirement 1: Please ensure that your manuscript meets PLOS ONE's style requirements, including those for file naming.

Authors’ response: We have followed PLoS ONE’s style requirements, including file naming conventions, in preparing the manuscript.

Requirement 2: We suggest you thoroughly copyedit your manuscript for language usage, spelling, and grammar. If you do not know anyone who can help you do this, you may wish to consider employing a professional scientific editing service. 

Authors’ response: As we also indicated for the editor’s comment below (comment 1), the manuscript has been edited carefully and extensively for language. The language is edited by a colleague, namely Yigerem Badego, who is an assistant professor of English language at Hawassa University, Ethiopia. The authors have also made their utmost effort to make sure that all problems with language (grammar and mechanics) are fixed. 

Requirement 3: Please provide additional details regarding participant consent. In the ethics statement in the Methods and online submission information, please ensure that you have specified (1) whether consent was informed and (2) what type you obtained (for instance, written or verbal). If your study included minors, state whether you obtained consent from parents or guardians. If the need for consent was waived by the ethics committee, please include this information.

Authors’ response: As we also responded to the editor’s comment below (comment 16), the ethics statement has been updated as per the comment. All required details are provided.

Requirement 4: Please include additional information regarding the survey or questionnaire used in the study and ensure that you have provided sufficient details that others could replicate the analyses. For instance, if you developed a questionnaire as part of this study and it is not under a copyright more restrictive than CC-BY, please include a copy, in both the original language and English, as Supporting Information.

Authors’ response: Additional information about the survey questionnaire has been provided under the “Data collection tools and procedures” part of the Methods and Material section. Besides, the questionnaires have been provided as supporting information (S1 & S2).

Requirement 5: In your Methods section, please provide additional information on how monthly income was categorised.

Authors’ response: As also indicated for the editor’s comment below (comment 15), we have provided additional information on how income was categorized in the Methods and Materials section, under “Data analysis procedure”.

Requirement 6: We noticed you have some minor occurrence of overlapping text with the following previous publication(s), which needs to be addressed:

- Hall, John E. Guyton and Hall textbook of medical physiology e-Book. Elsevier Health Sciences, 2015. The text that needs to be addressed involves some sentences of the Discussion. In your revision ensure you cite all your sources (including your own works), and quote or rephrase any duplicated text outside the methods section. Further consideration is dependent on these concerns being addressed.

Authors’ response: Now we have made sure that no text overlapping exists with the source mentioned. In some places in the manuscript (paragraph 9 of the discussion), we have changed our arguments and our references as the argument made based on the mentioned book (Guyton and Hall textbook of medical physiology) was not plausible. In other places where the mentioned book is cited (paragraph 8 of the discussion), we have made sure that there is no text overlap and a part of a sentence taken verbatim from the mentioned source in order to make our argument has been quoted.

Requirement 7: We note that you have stated that you will provide repository information for your data at acceptance. Should your manuscript be accepted for publication, we will hold it until you provide the relevant accession numbers or DOIs necessary to access your data. If you wish to make changes to your Data Availability statement, please describe these changes in your cover letter and we will update your Data Availability statement to reflect the information you provide. 

Authors’ response: Now we have provided the full dataset on which the manuscript is based (supporting information S3). We also have updated the data availability statement on the manuscript submission system by indicating that all data underlying the manuscript have been provided. 

 

Editor’s comments

Comment 1: There are many grammatical errors. These MUST all be corrected before re-submitting the manuscript. Otherwise the manuscript will not be fit for publication in PLOSONE.

Authors’ response: Thank you for pointing this out and for fixing some of the typographic and grammatical errors we committed. The language has now been carefully and extensively edited by the authors and by a colleague, namely Yigerem Badego, who is an assistant professor of English language at Hawassa University.

Comment 2: There are a number of additional statistical analyses that must be completed and results appropriately presented.

Authors’ response: Thanks for this suggestion. As elucidated in the point-by-point response below (comments 8, 9, 10 & 13), all required analyses/re-analyses have been done/re-done and the manuscript is updated accordingly.

Comment 3: The research data should be submitted with the revised copy of the manuscript.

Authors’ response: The full dataset on which the manuscript is based is now submitted with the revised version of the manuscript.

Comment 4: Please follow the author guidelines in writing the declaration section of the manuscript.

Authors’ response: Thank you for raising this point. We have followed the author guidelines in preparing the manuscript, including the declarations. Further, information that must be provided only in the submission system and that must not be included in the manuscript has been provided accordingly. 

Comment 5: The logistic regression needs to be described in more detail and the results of the odds rations appropriately interpreted.

Authors’ response: Thank you for this comment. As also indicated below (comment 7), we have provided further and more detailed description of the logistic regression in the “Data analysis procedure” part of the Methods and Materials section. Interpretation of the odds ratios has also been revised as per the comment. 

Comment 6: Provide details of specifications and assumptions of these sample size calculations?

Authors’ response: Dear editor, thank you for your comment. We now have provided details of all required assumptions and specifications pertaining to the sample size calculation.

Comment 7: You need to provide a detailed description [of] how the logistic model was built. This is very important to assess the usefulness of the model results. Therefore, the investigators need to do a good job of describing it. 

 Authors’ response: Thank you for bringing this issue to our attention. As per the comment, we have provided a detailed description of the model fitting and assessment of goodness-of-fit procedures in the “Data analysis procedure” part of the Methods and Materials section.

Comment 8: Was confounding assessed? If yes, how? If no, why not?

Authors’ response: We want to thank you for this important comment. Multivariable logistic regression model was used to handle covariates (and also confounders). We used the adjusted odds ratio, which is a confounder-adjusted measure of association, when investigating the risk factors associated with hypertension. Significance of association was evaluated based on the adjusted odds ratio. 

Comment 9: Was effect modification assessed? If yes, how? If no, why not?

Authors’ response: Thank you for this query. We assessed effect modification by including in the multivariable model possible and plausible interaction terms one at a time. We checked for interaction between BMI and age, BMI and sex, central obesity and sex, and khat chewing and alcohol drinking but none was found significant implying absence of effect modification. A description of how effect modification was assessed is now provided in the “Data analysis procedure” part of the Methods and Materials section. 

Comment 10: Check if this is normally distributed. If not, present median and interquartile range.

Authors’ response: We want to thank you for this important comment. We assessed the normal distribution for all the continuous variables in our study. We reported mean and standard deviation for normally distributed variables and median and interquartile range for positively or negatively skewed distributions. This is consistent throughout our paper. 

Comment 11: How do you define light smokers? Please add this to the paper.

Authors’ response: Thank you again. As per your comment and comments from Reviewer 1, we have clarified such terminologies and variables in the “Definition of variables” part of the Methods and Materials section. 

Comment 12: Recommendation of WHO. Explicitly state what this is.

Authors’ response: We want to thank you for this important comment. WHO experts have recommended fruit and vegetables intake on daily basis. Now we have removed the phrase “as per the recommendation of WHO” to avoid any ambiguity in the message being conveyed.

Comment 13: Test each of this [BMI & WHR] if they were normally distributed. If not, please present the appropriate stats.

Authors’ response: Thank you for this comment too. As we already stated earlier (comment 5), we have assessed normality for all the continuous variables in our study and reported results accordingly.

Comment 14: You are interpreting these as if they are relative risks. Please remember that these are AORs and should be interpreted as such. The differences are in odds NOT risks.

Authors’ response: Thank you. We have corrected the interpretation of our findings as per your comment (odds rather than risk) throughout the manuscript. 

Comment 15: In your Methods section, please provide additional information on how monthly income was categorized.

Authors’ response: We want to thank you for this comment. We now have provided a description of how income was categorized in the “Data analysis procedure” part of the manuscript.

Comment 16: Please provide additional details regarding participant consent. In the ethics statement in the Methods and online submission information, please ensure that you have specified (1) whether consent was informed and (2) what type you obtained (for instance, written or verbal). If your study included minors, state whether you obtained consent from parents or guardians. If the need for consent was waived by the ethics committee, please include this information.

Authors’ response: Thank you for the important comment. We have revised the ethics statement and provided additional details. No minors in our study since our study population was civil servants (adults).

---

## [Decision Letter · Decision Letter 1]

28 May 2020

Prevalence and risk factors of hypertension among civil servants in Sidama Zone, south Ethiopia

PONE-D-19-34743R1

Dear Dr. Badego,

We are pleased to inform you that your manuscript has been judged scientifically suitable for publication and will be formally accepted for publication once it complies with all outstanding technical requirements.

With kind regards,

Agricola Odoi, BVM, MSc, PhD, FAHA, FACE

Academic Editor

PLOS ONE

Additional Editor Comments (optional):

Reviewers' comments:

Reviewer's Responses to Questions

**Comments to the Author**

1. If the authors have adequately addressed your comments raised in a previous round of review and you feel that this manuscript is now acceptable for publication, you may indicate that here to bypass the “Comments to the Author” section, enter your conflict of interest statement in the “Confidential to Editor” section, and submit your "Accept" recommendation.

Reviewer #1: All comments have been addressed

Reviewer #2: All comments have been addressed

2. Is the manuscript technically sound, and do the data support the conclusions?

Reviewer #1: Yes

Reviewer #2: Yes

3. Has the statistical analysis been performed appropriately and rigorously? 

Reviewer #1: Yes

Reviewer #2: Yes

4. Have the authors made all data underlying the findings in their manuscript fully available?

Reviewer #1: Yes

Reviewer #2: Yes

5. Is the manuscript presented in an intelligible fashion and written in standard English?

Reviewer #1: Yes

Reviewer #2: Yes

6. Review Comments to the Author

Reviewer #1: I am satisfied with their responses to the points I raised in my review. I think it is ready to be published at this point. Good work in revision

Reviewer #2: This article explains a cross sectional study that was conducted to assess hypertension among civil servants in Ethiopia. There were many comments made by previous reviewers upon the initial submission of this article. These comments have been addressed and evidence of revisions were provided from the authors. There is a slight style revision that should be considered on line 34 AOR: 4.85 [95%CI] the authors should make this nomenclature consistent as it is different on the line above 33 e.g AOR 4.31.

7. PLOS authors have the option to publish the peer review history of their article (what does this mean?). If published, this will include your full peer review and any attached files.

Reviewer #1: Yes: John G. McPeak

Reviewer #2: No

---

## [Editor Report · Acceptance letter]

1 Jun 2020

PONE-D-19-34743R1 

Prevalence and risk factors of hypertension among civil servants in Sidama Zone, south Ethiopia 

Dear Dr. Badego:

I am pleased to inform you that your manuscript has been deemed suitable for publication in PLOS ONE. Congratulations! Your manuscript is now with our production department. 

With kind regards,

on behalf of

Prof. Agricola Odoi 

Academic Editor

PLOS ONE